# A qualitative exploration of the over-the-counter availability of oral contraceptive pills in Australia

**Zobaida Ahmed**[1,2] *, **Yuanyuan Gu**[1,2,3], **Kompal Sinha**[1,2], **Mutsa Mutowo**[1], **Natalie Gauld**[4], **Bonny Parkinson**[1,2,3]

**1** Macquarie University Centre for the Health Economy, Macquarie University, Sydney, NSW, Australia,
**2** Department of Economics, Macquarie Business School, Macquarie University, Sydney, NSW, Australia,
**3** Australian Institute of Health Innovation, Macquarie University, Sydney, NSW, Australia, **4** School of Pharmacy, University of Auckland, Auckland, New Zealand

\* Zobaida-ahmed.piu@students.mq.edu.au

**Data Availability Statement:** Data cannot be shared publicly because of ethical reasons. There are restrictions imposed by the Macquarie University Ethics Committee on sharing the de-

## Abstract

### Introduction

The prevention of unintended pregnancy is a public health issue affecting women worldwide. In Australia, women are required to get a prescription to obtain the oral contraceptive pill (OCP), which may limit access and be a barrier to its initiation and continuing use. Changing the availability of the OCP from prescription-only to over-the-counter (OTC) is one solution, however, to ensure success policymakers need to understand women's preferences. Telehealth services also might serve as an alternative to obtain prescriptions and increase accessibility to OCPs. This study aims to explore the preferences for OTC OCPs among Australian women, and whether the expansion of telehealth impacted women's preferences.

### Methods

A mixed methods approach was used to explore women's preferences regarding access to the OCP. Focus group discussions (FGDs) were conducted to organically identify the preferences followed by an empirical ranking exercise. Three FGDs in two phases were conducted, pre and post-expansion of telehealth in Australia due to the COVID-19 pandemic. Convenience sampling was employed. The technique of constant comparison was used for thematic analysis where transcripts were analysed iteratively, and codes were allowed to emerge during the process to give the best chance for the attributes to develop from the data.

### Results

Thematic analysis revealed that women perceived OTC availability of OCPs as a mechanism to increase the accessibility of contraception by reducing cost, travel time, waiting time, and increasing opening hours. They also believed that it would increase adherence to OCPs. However, some potential safety concerns and logistical issues were raised, including

identified transcripts because the focus group discussions contain sensitive topics, such as discussions on reproductive health and contraception, brand names of medicines, specific medical centres/pharmacies etc. The participants gave consent to "Only the investigators will have access to collected data. Data requests can be sent to: ethics.secretariat@mq.edu.au.

**Funding:** This research is funded by scholarships, including the Macquarie University Research Excellence Scholarship (20211229) and Macquarie University Center for Health Economy (MUCHE) top up scholarship (20203478), with support from Macquarie University and five pharmaceutical companies: Amgen Australia, Janssen Australia, MSD Australia, Pfizer Australia, and Roche Australia. The funders had no role in study design, data collection and analysis, decision to publish, or preparation of the manuscript.

**Competing interests:** This research is funded by scholarships, including the MQRES (No. 20211229) and MUCHE Top Up scholarships (No. 20203478), with support from Macquarie University and five pharmaceutical companies: Amgen Australia, Janssen Australia, MSD Australia, Pfizer Australia, and Roche Australia. The funders had no role in study design, data collection and analysis, decision to publish, or preparation of the manuscript. NG has provided consultancy services on reclassification, including regarding oral contraceptives, and led the New Zealand work to gain pharmacist-supply access to oral contraceptives. NG was a Board Member of the Pharmaceutical Society of New Zealand, which provides training on non-prescription oral contraceptives. This does not alter our adherence to PLOS ONE policies on sharing data and materials.

pharmacist training, access to patient's medical history, the ability to discuss other health issues or undertake opportunistic health screening, adherence to checklists, and privacy in the pharmacy environment. Following the expansion of telehealth, accessibility issues such as opening hours, travel time, and location of the facility were considered less important.

## Conclusions

The participants expressed their support for reclassifying OCPs to OTC, particularly for repeat prescriptions, as it would save valuable resources and time. However, some safety and logistical issues were raised. Women indicated they would balance these concerns with the benefits when deciding to use OTC OCPs. This could be explored using a discrete choice experiment. The expansion of telehealth was perceived to reduce barriers to accessing OCPs. The findings are likely to be informative for policymakers deciding whether to reclassify OCPs to OTC, and the concerns of women that need addressing to ensure the success of any policy change.

## 1 Introduction

The prevention of unintended pregnancy is a public health issue affecting women worldwide. Each year, approximately 121 million pregnancies worldwide are categorized as unintended, accounting for nearly half of all pregnancies [1]. An 'unintended pregnancy' is one that is mistimed, unplanned, or unwanted at the time of conception [2]. It can result from the use of less effective contraceptive methods, non-use of contraception, or instances of contraceptive failure, such as missed contraceptive pills or condom breakage.

In Australia unintended pregnancies cause a significant burden with almost 40% of pregnancies recorded as unintended in 2020, costing the economy $7.2 billion in direct and indirect economic costs [3]. According to an estimation of an Australian national survey, approximately one-third of women had an unplanned pregnancy at some point in their lives in 2013 [4]. Studies and reports highlight the higher rates of unintended pregnancies among Aboriginal and Torres Strait Islander (ATSI) women. One such study is the Australian Longitudinal Study on Women's Health, which reported that ATSI women are more likely to experience unintended pregnancies compared to non-ATSI women [5].

Contraception is an effective method of reducing unintended pregnancies. Although long-acting reversible contraception (LARC) agents are more effective [6] the most common form of contraceptive used by Australian women is the oral contraceptive pill (OCP), available as either a combination of oestrogen and progestogen or progesterone-only [7]. The effectiveness of OCPs in preventing pregnancy relies heavily on consistent adherence. Poor adherence and discontinuation of OCPs may result in unintended pregnancy [8].

Barriers to access have been cited as one of the major reasons for inconsistent contraceptive use by women [9]. Specifically for OCPs, the requirement for a prescription is a barrier to both initiating [10, 11] and continuing the use [12, 13]. Additionally, initial, and ongoing medical consultation costs for prescriptions are barriers [12, 14–16].

Removal of prescription requirements can improve contraceptive access, as occurred with the over-the-counter (OTC) availability of emergency contraception [17]. It may also encourage women to change from using less effective contraceptive methods to the OCP [18].

Furthermore, increasing access through OTC availability could reduce unplanned pregnancies and generate savings for the health system and society [18–20].

OCPs are available by prescription-only in Australia, although continued dispensing (4 months' supply) by pharmacists may be used in restricted situations [21]. Three states of Australia, namely Queensland, New South Wales, and Victoria, have announced state-wide pilots of OTC OCPs to provide more access [22–24]. Other countries have enabled pharmacists to supply hormonal contraception by various mechanisms. In the US there are different policies on how pharmacists in Washington DC and 20 other states can prescribe or supply hormonal contraception without a prescription [25], using: a collaborative practice agreement with a medical doctor; independent prescribing under state-wide protocols or through specific legislation; or state-wide standing orders [26–30]. In the UK, progestogen-only contraceptive pills were announced to be available in the pharmacies without any prescription in 2021. Women can access it for free of cost via general practitioners (GPs) or for a small fee at a pharmacy in the UK [31]. In New Zealand, specially trained pharmacists can prescribe non-prescription OCPs to women aged 16 years and above who have initially been prescribed OCPs by a GP [32]. Only an initial prescription is required in the Netherlands and repeats may be purchased from a pharmacy indefinitely thereafter [33].

Policymakers need to understand the preferences of consumers regarding reclassifying OCPs because the success of this policy depends on them. This study aims to explore the preferences for OTC OCPs among Australian women, including the potential benefit and negative consequences. It also explores the impact of an important policy change in Australia. Before 2020, telehealth was available for specialist care for patients located in telehealth eligible remote areas, ATSI medical services, and eligible aged care facilities. After March 2020 telehealth was expanded nationally and to GP services due to the COVID19 pandemic [34]. This paper explores whether the expansion of telehealth impacted women's preferences for OTC OCPs.

To date much of the literature on the public perception of OTC provision of contraceptives focused on emergency contraception pills in Australia [35–37]. However, emergency contraceptive pill attributes differ from the regular OCPs. Moreover, emergency contraceptive pills were rescheduled as a non-prescription OTC medicine in 2004 [38].

Studies conducted on the view of OTC availability of OCPs mostly focused on the USA. Baum et al. [39] examined the perspectives among a diverse sample of women on the possibility of obtaining OCPs OTC. Focus group discussions and in-depth interviews were conducted by Dennis and Grossman [40] in the Boston area which focused on exploring the barriers to contraception and interest in OTC OCP among low-income women. Potter et al. [14] examined the choices and motivations of OCP users living in El Paso regarding visiting a US clinic or a Mexican pharmacy with OTC OCPs. Barlassina [41] evaluated the views and attitudes of OCP users towards the availability of OCPs without a prescription in the Republic of Ireland. However, the findings from these studies cannot be directly applied to the Australian context because of Australia's unique healthcare system and healthcare financing strategy. Moreover, none of these studies had the scope to consider the impact of telehealth in primary healthcare as per the current study.

This study represents the first attempt, to the best of our knowledge, to investigate women's view on reclassification of regular OCP from 'prescription only' to OTC in Australia. Particularly, it considers both qualitative and quantitative approaches, allowing for a comprehensive analysis of women's views before and after the expansion of telehealth medicine. This distinctive focus on the intersection of OCP reclassification, telehealth, and women's opinions sets this study apart from previous research. The findings will be incorporated into a novel discrete choice experiment framework to explore how women prioritize and trade-off different attributes related to OCP accessibility.

## 2 Methods

### 2.1 Study design and research team

A mixed methods (qualitative and quantitative) approach incorporating focus group discussions (FGDs) and an empirical ranking exercise was used to explore women's preferences regarding OTC OCPs. We first conducted focus group discussions (FGD), followed by an empirical ranking exercise that included organically generated factors identified during the FGDs. The FGDs and ranking exercises were given equal priority when interpreting the results. This type of approach has been successfully employed and reported upon previously in several studies in healthcare context [42–45]. Using both qualitative and quantitative methods enable the generation and exploration of the rationale to identify important criteria for making decisions about healthcare resource allocation. In this process the views of the public are incorporated into the formulation of health policy [45]. FGDs were chosen as the method of data collection because they can spark memories in participants, and ideas and thoughts about issues can develop as they are discussed. Additionally, if some members of a group admit to feelings or thoughts that could be considered embarrassing or socially unacceptable, other members may find the confidence to share similar ideas or thoughts [46]. The Consolidated Criteria for Reporting Qualitative Research (COREQ) was utilized to report important aspects of the study methods, context of the study, analysis, findings and interpretation of the findings [47] (see S1 Appendix).

Three FGDs in two phases were conducted to explore factors that were important to women regarding OTC OCPs. The first phase of FGD (FGD1a and FGD1b) took place in December 2018. In 2020, prescriptions for OCPs could be obtained via telehealth consultations nationwide, which was not possible for most women pre-2020. Therefore, another phase of FGD (FGD2) was conducted in October 2022 focusing on OTC OCPs and whether women's preferences regarding the accessibility of OCPs had changed.

To be eligible, women had to speak English, be between the ages of 18 and 45 and currently not trying to conceive. Participants showed their interest via email. The FGD1a and FGD1b each comprised of seven women. The second phase of FGD (FGD2) comprised of eight women. A total of twenty-two women participated in the study through three FGDs. Convenience sampling was used for this study. Written consent was obtained from all participants. In both phases, participants were reimbursed for their time through the receipt of a gift card voucher. The study protocol was approved by Macquarie University Ethics Committee, Business and Economics Subcommittee 5201838305331 in 2018 and 520221202741750 in 2022.

### 2.2 Data collection

Two topic schedules were developed with reference to published literature on views of OTC OCPs [10, 48–50].

The topic schedule used for FDG1 (FGD1a, FGD1b) aimed to identify the important factors affecting contraception choices. Questions were asked about the knowledge on contraceptive methods available in Australia, factors considered when choosing a contraceptive method, and the sources of information they considered when choosing a contraceptive method. FGD2 topic schedule mainly focused on OTC OCPs and the expansion of nationwide telehealth consultation in the Australian primary healthcare system. In both topic schedules, participants were presented with a hypothetical scenario regarding accessing OCP from the pharmacist without the requirement of a prescription and asked for their opinions on this scenario. The factors which were perceived important from the discussions were listed together, and the participants were asked to rank the factors. The participants were asked to rank the most

important factor as 1, the second most as 2 and so on. The least important factor received the highest score ranking exercise. In both phases, participants were given a survey questionnaire collecting socio-demographic information, and their current and past use of contraceptives.

## 2.3 Data analyses

The technique of constant comparison was used for thematic analysis [51] where transcripts were analysed iteratively and codes were allowed to emerge during the process to give the best chance for the attributes to develop from the data. To begin defining the themes, codes (labels) were added to chunks of text. The analysis was more inductive than deductive allowing the flexibility to adapt to the data [52]. Following Dey [53] some codes were spliced or linked (e.g., the codes were either split into separate codes, or some were joined to create one code from a number of others). Ultimately, the aim was to consolidate low-level themes into higher-order themes. Themes were described in coding frames/descriptive accounts, where they were labelled and defined with examples given from the texts [52]. Quotes are presented for the identified themes in S2 Appendix. The ranking exercise guided the selection of the major and minor themes related to the OTC OCP, which is demonstrated in the mind map in S3 Appendix. NVivo-20 was used for analysing the data. From the FGDs, the conceptual mapping of the decision process was constructed (Fig 1).

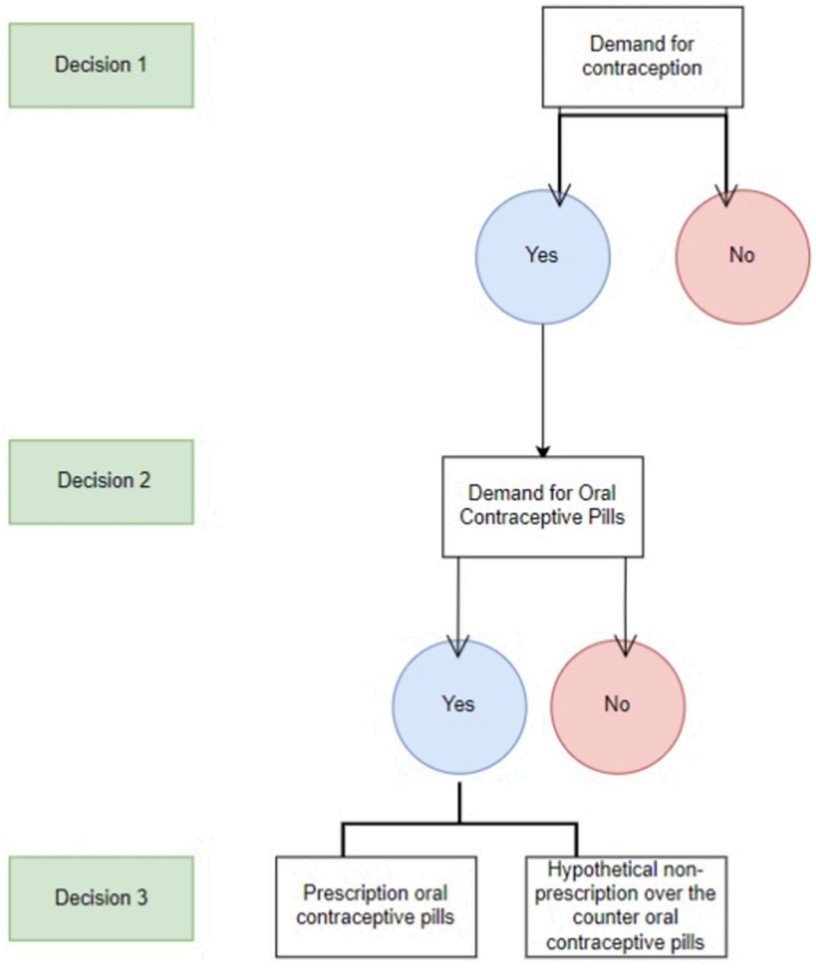

**Fig 1. Decision tree for OTC OCP.**

This study focused only on decision 3. There were some evident changes in the preferences for OTC OCPs from FGD1 to FGD2. However, no new code emerged during FGD2, which indicated data saturation had been reached on the topic.

# 3 Results

## 3.1 Respondent characteristics

Table 1 presents respondent characteristics, collected through questionnaires after each FGDs. Participants were similar in age, with the median age being 30–34 years. Most participants were never married (FDG1: 64.3%, FDG2: 62.5%) and had a university degree (FDG1: 78.5%, FDG2: 72.5%). There were differences in terms of full-time employment (FGD1: 64.3%, FDG2: 25%) and whether they were born in Australia (FDG1: 50.0%; FDG2: 25.0%).

**Table 1. Sociodemographic characteristics of women participating in focus group discussions.**

| | FGD1 | FGD2 |
|---|---|---|
| **Characteristics** | **N (%)** | N (%) |
| **Age** | | |
| **18–24 years** | 2 (14.3) | 2 (25.0) |
| **25–29 years** | 2 (14.3) | 1 (12.5) |
| **30–34 years** | 6 (42.9) | 3 (37.5) |
| **35–39 years** | 2 (14.3) | 2 (25.0) |
| **Over 40 years** | 2 (14.3) | 0 (0.0) |
| **Relationship status** | | |
| **Never married** | 9 (64.3) | 5 (62.5) |
| **Widowed** | 1 (7.1) | 0 (0.0) |
| **Divorced** | 0 (0.0) | 0 (0.0) |
| **Separated but not divorced** | 0 (0.0) | 0 (0.0) |
| **Married** | 4 (28.6) | 3 (37.5) |
| **Level of highest educational attainment** | | |
| **Postgraduate certificate, diploma or degree** | 3 (21.4) | 4 (50.0) |
| **Bachelor degree** | 8 (57.1) | 1 (12.5) |
| **Certificate, advanced diploma/diploma** | 3 (21.4) | 2 (25.0) |
| **Year 12 or below** | 0 (0.0) | 1 (12.5) |
| **Level not determined** | 0 (0.0) | 0 (0.0) |
| **Birth country** | | |
| **Australia** | 7 (50.0) | 2 (25.0) |
| **Other** | 7 (50.0) | 6 (75.0) |
| **Language spoken at home** | | |
| **English** | 12 (85.7) | 5 (62.5) |
| **Other** | 3 (21.4) | 3 (37.5) |
| **Employment** | | |
| **Employed (full time)** | 9 (64.3) | 2 (25.0) |
| **Employed (part-time)** | 2 (14.3) | 1 (12.5) |
| **Employed (casual)** | 1 (7.1) | 0 (0.0) |
| **Student** | 3 (21.4) | 3 (37.5) |
| **Unemployed (looking for job)** | 0 (0.0) | 1 (12.5) |
| **Unemployed (home duty)** | 0 (0.0) | 1(12.5) |
| **Religion specified** | 8 (57.1) | 4 (50.0) |
| **No religion** | 6 (42.9) | 4 (50.0) |
| **All participants** | **14 (100.0)** | 8 (100.0) |

**Table 2. Experience with contraceptives by women participating in focus group discussions.**

| | N (%) | | | |
|---|---|---|---|---|
| | Ever used contraceptives FGD1 | Ever used contraceptives FGD2 | Current contraception use FGD1 | Current contraception use FGD2 |
| **Contraceptive method** | | | | |
| **Not using contraception (trying to conceive)** | 2 (14.3) | 1(1.25) | 0 (0.0) | 0(0.0) |
| **Not using contraception (not trying to conceive)** | 2 (14.3) | 1(1.25) | 2 (14.3) | 2 (25.0) |
| **Condom only (male or female)** | 7 (50.0) | 5 (62.5) | 2 (14.3) | 2 (25.0) |
| **Condom + implant** | 1 (7.1) | 0 (0.0) | 1 (7.1) | 0 (0.0) |
| **Condom + IUD*** | 1 (7.1) | 0 (0.0) | 0 (0.0) | 0 (0.0) |
| **Condom + injection** | 2 (14.3) | 0 (0.0) | 1 (7.1) | 0 (0.0) |
| **Condom + OCP** | 2 (14.3) | 3 (37.5) | 2 (14.3) | 0 (0.0) |
| **Condom + other (please specify)** | 0 (0.0) | 1 (12.5) | 0 (0.0) | 0 (0.0) |
| **Condom + withdrawal** | 3 (21.4) | 0 (0.0) | 1 (7.1) | 0 (0.0) |
| **Implant** | 1 (7.1) | 0 (0.0) | 1 (7.1) | 0 (0.0) |
| **IUD** | 1 (7.1) | 1 (12.5) | 0 (0.0) | 1 (12.5) |
| **Injection (e.g. Depo-Provera)** | 1 (7.1) | 0 (0.0) | 0 (0.0) | 0 (0.0) |
| **OCP** | 9 (64.3) | 6 (75.0) | 7 (50.0) | 4 (50.0) |
| **The mini pill (progestogen only)** | 1 (7.1) | 2 (2.22) | 0 (0.0) | 0 (0.0) |
| **Withdrawal** | 3 (21.4) | 1 (12.5) | 0 (0.0) | 0 (0.0) |
| **Natural (safe period) methods** | 2 (14.3) | 2 (25.0) | 0 (0.0) | 0 (0.0) |
| **Vaginal ring** | 0 (0.0) | 1 (12.5) | 0 (0.0) | 0 (0.0) |
| **Emergency contraceptive pill** | 5 (35.7) | 2 (25.0) | 0 (0.0) | 0 (0.0) |
| **Female sterilization** | 0 (0.0) | 0 (0.0) | 0 (0.0) | 0 (0.0) |
| **Male sterilization** | 1 (7.1) | 0 (0.0) | 0 (0.0) | 0 (0.0) |
| **Reasons for using contraception** | | | | |
| **Preventing pregnancy** | 12 (85.7) | 7 (87.5) | | |
| **Regulating period** | 10 (71.4) | 2 (25.0) | | |
| **Reducing cramps/pain** | 6 (42.9) | 2 (25.0) | | |
| **Preventing infectious diseases** | 2 (14.3) | 0 (0) | | |
| **Treating acne** | 4 (28.6) | 3 (37.5) | | |
| **Treating endometriosis** | 2 (14.3) | 0 (0) | | |
| **Other** | 1 (7.1) | 0 (0) | | |
| **Experience of side effects** | | | | |
| **Yes** | 10 (71.4) | 4 (50.0) | | |
| **No** | 4 (28.6) | 3 (37.5) | | |
| **Unsure** | 0 (0.0) | 1 (12.5) | | |
| **Total** | **14 (100.0)** | 8 (100.0) | **14 (100.0)** | 8 (100.0) |

IUD = Intra Uterine Device

Table 2 shows that participants in both FGD phases were similar in terms of contraceptive methods used–most had ever used the OCP alone (FGD1: 64.3%, FGD2: 75%), followed by condoms (FGD1: 50.0%, FGD2: 62.5%), or a combination of both (FGD1: 14.3%, FGD2: 37.5%). In both FGDs 50.0% participants were currently using only OCPs as the primary method of contraception. Many participants had previously used higher-risk contraceptive methods, such as withdrawal (FGD1: 21.4%, FGD2: 12.5%) or natural/safe period methods (13.3% in FGD1 and 25% in FGD2). Fewer had ever used (FGD1: 21.3%, FGD2:12.5%) or

were currently using (FGD1: 7.1%, FGD2: 12.5%) LARCs (e.g. IUD, injection or implants). Some participants were current contraceptive non-users (FGD1: 14.3%, FDG2: 25.5%).

Most respondents reported that preventing pregnancy was a key reason for using contraception (FGD1: 85.7%, FGD2: 87.5%), followed by regulating the period (71.4% in FGD1), treating acne (37.5% in FGD2), and reducing period cramps or pain (FGD1: 42.9%, FGD2: 25.0%). Most participants had previously experienced side-effects (FGD1: 71.4%, FGD2: 50.0%).

### 3.2 Focus group discussion findings

FGD1 began with discussing initial thoughts regarding factors affecting their choice of contraception. Factors mentioned included: effectiveness in preventing pregnancy, side effects, mode and frequency of administration, ease of use, ability of regulating period and treatment for other health issues.

*"For me, the Pill is the best. I thought that Implanon might be because you don't have to worry about it, you don't have to remember to take it and it's kind of just dealt with, you don't have to think about it. But because I had side effects on that and weeks of bleeding and things like that.." FGD1a, ID-5*

Following the two phases of focus groups discussion, six major themes regarding OTC OCPs were identified: accessibility, costs, privacy, trustworthiness, opportunistic health screening, and safety issues.

**3.2.1 Accessibility and convenience.** The participants thought the requirement for a prescription decreased accessibility and adherence to OCPs.

*"I know I've skipped it for a week at a time if I couldn't get to the doctor." FGD1, ID-2*

The participants agreed that pharmacy availability of the OCPs would increase accessibility for young women and women with a lower socioeconomic status.

*"Yeah, or like you said, are still dependent on their parents for Medicare and stuff. They should have access to contraceptives when they want, whichever the want." FGD1a, ID-7*

This view for younger women remained unchanged in FGD2.

*"I would say like kind of from what we've been saying, like it definitely would just be more convenient, especially say if you are a younger female. And that is just more accessible. . ." FGD2, ID-20*

Some respondents thought that the OTC OCPs would increase accessibility for women with a lower socio-economic status by reducing the GP consultation cost. It was also considered beneficial for women in remote areas with limited access to GPs.

*"I think it would increase, particularly for like low socio-economic areas where they might not have easier access to doctors. . ." FGD1a, ID-4*

Participants considered pharmacies to be more flexible than medical centres in terms of opening hours and thus OTC OCPs would increase accessibility for employed women.

"*I just think sometimes you can't always get a doctor's appointment or it's really difficult with work and work hours and so I wish my doctor would let us just call. But it can take like a good week sometimes just to sort it out.*" FGD1b, ID-6

In contrast, FGD2 placed less importance on opening hours, possibly due to the option of telehealth consultations with GPs. The availability of electronic prescription and online purchase with delivery might have changed participant preferences regarding opening hours as a constraint.

"*But then again, you can get it shipped to you. The prescriptions. Sometimes you'll like a pharmacy. They do like a drive-around, they do delivery for like five bucks or something.*" FGD2, ID-16

Some women believed that OTC OCPs would increase the contraceptive options for women.

"*So it just gives everyone more options really, which is a good thing.*" FGD1b, ID-8

Many participants had good GP access, although some had difficulties due to distance compared to pharmacies.

"*My GP's further away from my house than the chemist.*" FGD1a, ID-2

The importance of waiting time and consultation time was evident from the spontaneous discussion of the participants in FGD1. OTC OCPs were considered to avoid the waiting time for GPs.

"*Doctors take forever. My old GP I'd had since I was a kid, he was the best. He would see me sitting there and would be like—called my name ahead and call me in within half an hour. So, I didn't mind so much them. He retired and I [now] have to wait two hours to see a GP. It's disgusting.*" FGD1a, ID-3

A long waiting time and a short consultation time to fill an OCP prescription were considered wasteful by some participants.

"*You're there for five minutes and you're like, I just need a script and then you like, go.*" FGD1b, ID-10

The waiting time to see a GP was less important in FGD2 for some participants and it was one of the least important factors to them. The opportunity of telehealth consultation enabled GP consultations without being physically present.

"*It depends if I'm going in-store or getting it online, I guess. I mean if it's something that I really need–I'll–right now–you know what I mean? Like if it's a really pressing issue. Otherwise, no.*" FGD2, ID-16

In contrast, some participants were sceptical about the duration of the consultation time with the pharmacists with OTC OCPs given the busy nature of pharmacies.

"*Look when I'm honest now, with the pharmacy fill out the—normally it's—they're very busy do don't have much time to talk to you and basically put it in so much in consideration, so it's*

*basically just give to me and I have to, somebody else standing already behind me...” FGD1b, ID-12*

Women believed that it would be easier to stay on the pill longer if it were available OTC due to easy access, which echoed the voice of FGD1 participants who viewed the requirement for a prescription reduced accessibility and adherence to OCPs.

“*There could be less hassle for the people for the long term use*” FGD2. ID-22

Some participants from FGD2 considered OTC OCPs may divert women from other contraceptive options, like LARCs.

“*It might bias people towards the oral contraceptive pill when an IUD or something else may be more appropriate. So accessibility may drive people to that being the more obvious choice.*” FGD2, ID-15

The participants of FGD2 were more flexible about the accessibility issues like waiting time, opening hours and travel time than FGD1 participants. It indicated that access to telehealth consultation changed women's perception of accessibility factors.

**3.2.2 GP consultation costs.** Most participants expected OCP OTC would be cost-saving due to reduced GP consultation costs. This was considered especially applicable for women who have high out-of-pocket GP consultation costs. In Australia, most GPs provide bulk billing, which is a practice of choosing to be paid directly by the government leaving no out-of-pocket costs for the patient. However, many GPs are reducing bulk-billing [54]. So, consultation costs were a major concern for participants. When participants were asked about the major benefits of OTC OCPs, they said:

“*I think more GPs are moving away from bulk billing. So you know, if you have a GP that doesn't bulk bill, you'll get in a repeat prescription, let's say. It's an expense. If it's the first, again, the first time. Maybe it's a suitable model. If it's not, again, that's another hurdle, if you're paying 70 dollars to see your GP.*” FGD2, ID-15

**3.2.3 Privacy and confidentiality.** A woman's privacy when obtaining OCPs was emphasised from the very beginning of the discussions. There were mixed views about protecting privacy in a GP setting versus a pharmacy setting. First, participants were concerned about maintaining a young woman's confidentiality with GPs, whereas they thought pharmacists would not disclose their information to others.

“*I think that's a big one, especially with certain families and certain cultures, especially if you see a family GP. If you can do that without anybody knowing.*” FGD1a, ID-5

Participants also expressed the feeling that women did not like contraceptive use to be recorded. Therefore, it may be considered beneficial if OCPs were available without prescription.

“*I personally would not mind but I know people who have opted out already so they wouldn't want such information to go...*”FGD1a, ID-7

Second, participants were concerned about privacy in the pharmacy environment in terms of discussions being overheard by people waiting in the queue if there was a lack of a private room.

"*Because only for the fact that if they're going to ask me, for example, I'm standing there and they're like, so are you sexually active at the moment? How many sexual partners have you had this month, you know, those sorts of things..*" FGD1a, ID-1

Violation of privacy in the pharmacy setting was reiterated in FGD2.

"*That's even true- there is no other like separate area where they're going to give the medication and give the information. . . .*" FGD2, ID-16

**3.2.4 Trustworthiness toward provider.** Participants considered that pharmacists have less knowledge about a patient's medical history than GPs and were concerned that a woman might forget to tell them something, affecting the pharmacist's knowledge to help them provide contraceptives safely.

"*Like you said, GP knows you better. The pharmacist might not know what else is going on with your body. So that could be a disadvantage.*" FGD1a, ID-4

The participants from FGD2 were not as concerned about the unavailability of medical history to the pharmacist with OTC OCPs.

"*I mean you can get a printout of that. You can ask him for a summary. But again, as a child, like I wouldn't know to do that. I know to do that now. You can just be like, "Hey, can you just print out a summary of all my problems?" And then I'd take that to the pharmacist and be like, "Hey, this is all the stuff that I'm on."* FGD2, ID-16

Participants also considered the high turnover rate of pharmacists and lack of accountability or availability for follow-up, in contrast to a regular GP.

"*But okay, if we come back to pharmacists, if something, you've got a negative or something, you have an allergy, how pharmacist know about this? You have problem, you come back to pharmacist, say excuse me, you [didn't know] what are going to do, be complaining*" FGD1b, ID-7

Some participants raised concerns over pharmacists' training and medical knowledge.

"*Look I just trust the doctor more. I mean I trust doctor more because I think he's more educated in the ways that he learns more than pharmacist. Pharmacist is a person who know about drugs, doctor have to have more than that.*" FGD1b, ID-14

Participants also stated that pharmacists could be influenced by pharmaceutical company representatives.

"*I just thought of something. One pharmacy prescribed one particular pill than another, you know what I mean, could they be influenced by the pharma companies to. . .*" FGD1b, ID-9

FGD2 participants also feared that depending on the pharmacists for prescribing OCPs would promote expensive brands.

"*And like even to say like the one about pharmacists promotes the most expensive brand. Like in terms of trustworthiness, I would trust that they wouldn't.*" FGD2, ID-19

**3.2.5 Opportunistic health screening.** GPs often take the opportunity to discuss other health related issues and undertake opportunistic health screening, such as cervical screening and STI tests, when prescribing an OCP. Participants believed that OTC OCPs would reduce the opportunity to have regular health check-ups with GPs.

*I would agree though with that, it's like if you did go, like if you were at the doctor's and they said they mentioned do you want to get a check-up, then you could do it right then and there, whereas if you're a pharmacy and they kind of remind you, oh yeah, maybe I should, but it's like an inconvenience to then go to the doctor. So you probably keep at the back of your head, but get distracted by life and you don't actually get it done because it's like that added step.* FGD1b, ID-10

Some participants thought that OTC OCPs would reduce the probability of cervical screening.

"*Oh, that's a good question. Because usually when you go for your script, the first thing they ask you is when's your last pap smear? That does prompt that question..*"FGD1a, ID-6

Some women in the present study were concerned that OTC OCPs may reduce contact with the GP which might result in increased STIs.

"*It's just not a hormonal pill. I think the other—coming back again, the disadvantage again is STIs. So if you go to a GP and you get a referral, they are likely every now and then to say, "Have you had a screening?" So if you remove that, you might end up with a situation where people are knowing carrying STIs around with them.*" FGD2, ID-15

They considered the lack of a health check-up by a GP consultation to be wasteful. The ability to have other health check-up is considered as one of the major motives for visiting a GP to receive a prescription for OCPs.

**3.2.6 Safety issues.** Women were concerned about safety issues with OCPs. The participants did not consider a checklist for pharmacist to be a useful tool for assessment. Participants from all FGDs also noted that the emergency contraceptive pill checklist was not appropriately followed in their experience.

"*Another thing about that sort of checklist idea, like I know like my local 24-hour pharmacy on High Street in my suburb, me and my friends like, you know, you've all been there getting Plan B at one point or another. And some of them are like, "Yes, I didn't get a survey. I just asked the man for it and he gave it to me." Like I don't think it will be implemented. . ..*" FGD2, ID-21

The participants were concerned that OTC OCPs would encourage teenagers to have sex earlier.

"*Facilitator: all right, let's talk about the disadvantages of the pharmacist providing the pill without a prescription.*

*Participant: Kids having sex*" FGD1b-ID8

Some participants were worried about the possibility of abuse of OTC OCPs through overuse.

"*I think that I agree with her contribution is like at the sense of like having, you know, the record of if a person is abusing it. So when it comes through the prescription, I think they have a record of that if the person did get the, you know, contraceptive for any reasons previously..*" FGD2, ID-22

Participants also discussed abuse in the relationship. They considered that women might lose the opportunity to discuss about their abusive relationship if they do not have to see GP for getting prescription. One participant mentioned:

"*When I think abuse I maybe think of people who are in like abusive relationships, and then– or like maybe the age thing and at least with the GP you have to go through someone and maybe discuss things. But if you're able to just grab it and go, there's no sort of buffer for that*" FGD2 ID-20

There was unanimous consensus among participants that providing OTC OCP repeats to women who were already using the pill was safe and acceptable for all ages, including teens. However, they believed the initial prescription should come from the GP after conducting proper health assessments.

"*I think you need to see a doctor and get your blood, your blood pressure checked and everything before you go on the pill. And your weight and everything. Because it is important. But I think if you're getting re-prescribed something and you not bring up issues, like the repeats I don't think–I think that you should just be able to do that via telehealth consultation*" FGD2, ID-19

Whether online (text-based) pharmacy consultation was a valued addition to the existing system was further explored in FGD2. The participants compared this option with OTC availability of OCPs because they cannot see the GPs in person, neither any health assessment takes place in that mode.

"*I'm not sure what the value added would be there. That to me again seems like a consultation that isn't assessing me physically. Like again with the telehealth. And yes, someone's asking me questions, but I would wonder why that couldn't be done by a pharmacist who also does provide that service in a chemist.*" FGD2, ID-1

The participants believed receiving a repeat prescription for OCPs through an online-based consultation was just saving time and money without any additional value.

### 3.3 Ranking results

Table 3 presents the 10 most important factors found from the ranking exercise of FGD1 and FGD2. In FGD1, the effectiveness on preventing pregnancy was considered the most

**Table 3. List of 10 most important factors from ranking exercise.**

| | Ranking from FGD1 | Ranking from FGD2 |
|---|---|---|
| **Rank** | **Factors perceived important for selecting OCP as a contraception** | **Factors perceived important while accessing OCP** |
| 1 | Effectiveness of preventing pregnancy | Cost |
| 2 | Effect of pill on period | Privacy during consultation |
| 3 | Requirement of prescription | Trustworthiness |
| 4 | Location of GP | Confidentiality from others |
| 5 | Numbers of pills prescribed | Ensuring long term use safety |
| 6 | Cost of the pill | Ability to discuss other health issues |
| 7 | Effect on mood | Provider's behaviour |
| 8 | Type of administration | Stigma from the providers about OCPs |
| 9 | Waiting time to be seen by GP | Consultation time |
| 10 | Effect of pill on weight | Location of provider |

important factor influencing the choice of contraceptive method, followed by the effect on periods. The requirement for prescription was the third most important factor was also reflected in the FGD1 and FGD2 discussions. If only the accessibility issues were considered from the FGD1 ranking exercise, the requirement for prescription, location of the provider, cost, and waiting time were the most important factors.

FGD2 ranking exercise revealed the factors which were perceived important while accessing OCP. Cost was the most important factor, followed by privacy during consultation, trustworthiness, and confidentiality from others. From Table 3 it is evident that, the accessibility issues such as requirement of prescription, location of the provider, and waiting time were no longer important in FGD2 whereas to the participants of FGD1, these factors were on the list of top 10.

## 4 Discussion

This study identified the benefits and concerns regarding making the OCP available OTC held by focus group participants. The primary advantage of removing the requirement for a prescription included saved time from travelling to, waiting, and consulting with GPs, and saved GP consultation costs. These accessibility issues were also reported in previous studies [10, 40, 49, 55]. Generally, participants considered OTC availability as a convenient way to access OCPs. Participant opinions were consistent with past research involving reproductive health providers, pharmacists, and patients, that found increased access to be the primary benefit [10, 56, 57]. Although participants in the second phase (FGD2) perceived the accessibility issues such as waiting time and travel time less important than the first phase (FGD1). The expansion of nationwide telehealth services is likely to be the reason for this change in perceptions as it reduces time to access healthcare and increases patient satisfaction [58, 59].

The participants perceived the cost of OTC OCP to be cheaper due to avoided GP consultation fee costs. Most participants felt that this convenient, cheaper option would be an advantage for women from lower socioeconomic status and remote areas. This finding is consistent with Landau et al. [10]. Potter et al. [15] also found that women frequently mentioned "cost saving- as a means of not having to go to a doctor for a prescription" as the predominant rationale for choosing OTC OCP.

Women in the current study also thought it would facilitate their ability to maintain long-term OCP usage, consistent with the findings of a US cohort study [12]. Although the participants were mostly positive about the expanded role of the pharmacists and increased

accessibility and usage, they were also concerned about safety. They were worried that offering easy access to the OCP would encourage sex at an earlier age. A similar concern was raised by the respondents in Grossman et al. [49]. However, this concern is common for all other contraceptive options like condoms and emergency contraceptive pills, which are already available OTC. The experience of allowing emergency contraception to be available OTC found that it did not promote sexual risk taking among adolescents [55]. Other concerns about young women and possibility of contraindications were also not supported by evidence. Evidently, younger women are significantly less likely than older women to have contraindications to combined OCPs [60].

Most participants felt that having accurate and complete information about OCP use and side-effects was a key factor in a woman's ability to use it correctly and consistently. Participants also agreed that there should be an initial prescription from the GP to initiate OCPs, then there can be repeats from the pharmacists. This finding is similar to Baum et al. [39]. From their experience, some participants believed that a checklist was not followed properly by pharmacists. More research is needed in the Australian context regarding following checklists in pharmacies because women with known contraindications for combined OCPs might be at increased risk [61].

Some participants feared that the OTC availability of OCPs would discourage women to obtain recommended preventive services from GPs. However, US evidence suggests that the large majority of women continue to see their healthcare provider for these kinds of tests whether or not they are using a prescription contraceptive [10]. Another concern was raised that increased accessibility of OCPs would reduce use of more effective contraception, such as LARCs. Nonetheless, Coombe et al. [62] found that discomfort and myths around LARCs were the responsible factors for non-use of these effective methods. Furthermore, LARCs were shown to be only considered for use after dissatisfaction with shorter term methods such as OCPs. It is the contraceptive characteristics that is to be blamed for the low use of LARCs, not the accessibility of any method. This was also evident in the discussion of advantages of OCPs by the participants.

The participants valued the guidance and information they received from their GPs. They expressed their intention to continue seeking GP's advice even if OCPs are available OTC. Concerns about privacy and confidentiality at both pharmacies and GPs were noted by the participants. These findings were supported in previous studies [48, 49]. Efforts to create spaces that assure confidentiality are vital in implementation efforts if access is truly to be improved. Furthermore, securing greater privacy protections for individuals dependent on parents' Medicare card (refers to the identification card issued to individuals who are enrolled in the government-funded healthcare program known as Medicare) is an important factor to include for viable pharmacist prescribing.

Participants from phase 2 believed that the accessibility barriers of OCP were mostly reduced by the ability to have telehealth consultations and electronic prescriptions. With the expansion of nationwide telehealth, numerous accessibility issues such as opening hours, travel time, and location of the facility were considered less important. This is the most notable distinction between the participants of phase 1 and phase 2. It is also increasingly recognised in other studies that telehealth services reduce wait times and increase patient satisfaction [58, 59, 63]. Prior to the expansion of telehealth service, all these accessibility issues were emphasized, which were accompanied by a strong focus on cost related concerns. However, the cost of accessing OCPs was mentioned as the most important factor in phase 2 and they were ready to compromise on other accessibility aspects. Future research and advocacy efforts should explore if removing the access barriers through telehealth lowers the cost.

### 4.1 Limitations of the study

The relatively small sample of women included in this study were largely socio-demographically homogeneous and was not representative of the broader Australian population, limiting the generalisability of our findings. However, this sample was not intended to be representative, but rather to recruit a group of women able to provide insight into the research question posed. Second, due to limitations in the study protocol, women younger than 18 years old were not interviewed. Their perspective was discussed from an older person's point of view. Thus, it is possible that interviews would have produced different themes if younger teens were also included.

## 5 Conclusion

The participants in this study expressed their support for reclassifying OCPs to OTC especially for repeat prescriptions, as it would save valuable resources and time. They also believed that it would increase adherence to OCPs. However, some potential safety concerns and logistical issues were raised. Women would balance these concerns and issues with the benefits when deciding to use OTC OCPs. This could be explored in future research involving a discrete choice experiment approach. The expansion of nationwide reimbursed telehealth GP consultations was also perceived to reduce barriers to access OCPs. The findings are likely to be informative for policy makers deciding whether to reclassify OCPs to OTC, and the concerns of women that need to be addressed to ensure the success of a change in policy.

## Supporting information

**S1 Appendix. COREQ checklist.**
(DOCX)

**S2 Appendix. Major and minor themes with responses from the participants.**
(DOCX)

**S3 Appendix. Mind map for over-the-counter accessibility.**
(DOCX)

## Acknowledgments

The authors would like to thank the women who participated in the focus group discussions, as well as Mona Aghdaee and Emma Olin for their research assistance.

## Author Contributions

**Conceptualization:** Mutsa Mutowo, Natalie Gauld, Bonny Parkinson.

**Formal analysis:** Zobaida Ahmed.

**Methodology:** Bonny Parkinson.

**Project administration:** Bonny Parkinson.

**Supervision:** Yuanyuan Gu, Kompal Sinha, Bonny Parkinson.

**Visualization:** Zobaida Ahmed.

**Writing – original draft:** Zobaida Ahmed.

**Writing – review & editing:** Zobaida Ahmed, Yuanyuan Gu, Kompal Sinha, Mutsa Mutowo, Natalie Gauld, Bonny Parkinson.

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
