## [Decision Letter · Decision Letter 0]

16 Jan 2024

PONE-D-23-29682A qualitative exploration of the over-the-counter availability of oral contraceptive pills in AustraliaPLOS ONE

Dear Dr. Piu,

Thank you for submitting your manuscript to PLOS ONE. After careful consideration, we feel that it has merit but does not fully meet PLOS ONE’s publication criteria as it currently stands. Therefore, we invite you to submit a revised version of the manuscript that addresses the points raised during the review process.

We look forward to receiving your revised manuscript.

Kind regards,

Dr. Oluwatosin Olu-Abiodun

Academic Editor

PLOS ONE

Journal Requirements:

"This research is funded by scholarships, including the MQRES and MUCHE Top Up scholarships, with support from Macquarie University and five pharmaceutical companies: Amgen Australia, Janssen Australia, MSD Australia, Pfizer Australia, and Roche Australia."

Please state what role the funders took in the study.  If the funders had no role, please state: ""The funders had no role in study design, data collection and analysis, decision to publish, or preparation of the manuscript."" If this statement is not correct you must amend it as needed. 

4. Thank you for stating the following in the Financial Disclosure section: 

"This research is funded by scholarships, including the MQRES and MUCHE Top Up scholarships, with support from Macquarie University and five pharmaceutical companies: Amgen Australia, Janssen Australia, MSD Australia, Pfizer Australia, and Roche Australia."

We note that you received funding from a commercial source: Amgen Australia, Janssen Australia, MSD Australia, Pfizer Australia, and Roche Australia.

Within this Competing Interests Statement, please confirm that this does not alter your adherence to all PLOS ONE policies on sharing data and materials by including the following statement: ""This does not alter our adherence to PLOS ONE policies on sharing data and materials.” (as detailed online in our guide for authors http://journals.plos.org/plosone/s/competing-interests).  If there are restrictions on sharing of data and/or materials, please state these. Please note that we cannot proceed with consideration of your article until this information has been declared. 

"NG has provided consultancy services on reclassification, including regarding oral contraceptives, and led the New Zealand work to gain pharmacist-supply access to oral contraceptives. NG was a Board Member of the Pharmaceutical Society of New Zealand, which provides training on non-prescription oral contraceptives."

6. We note that you have indicated that there are restrictions to data sharing for this study. PLOS only allows data to be available upon request if there are legal or ethical restrictions on sharing data publicly. For more information on unacceptable data access restrictions, please see http://journals.plos.org/plosone/s/data-availability#loc-unacceptable-data-access-restrictions. 

7. PLOS requires an ORCID iD for the corresponding author in Editorial Manager on papers submitted after December 6th, 2016. Please ensure that you have an ORCID iD and that it is validated in Editorial Manager. To do this, go to ‘Update my Information’ (in the upper left-hand corner of the main menu), and click on the Fetch/Validate link next to the ORCID field. This will take you to the ORCID site and allow you to create a new iD or authenticate a pre-existing iD in Editorial Manager. Please see the following video for instructions on linking an ORCID iD to your Editorial Manager account: https://www.youtube.com/watch?v=_xcclfuvtxQ

9. We note that this data set consists of interview transcripts. Can you please confirm that all participants gave consent for interview transcript to be published?

If they DID provide consent for these transcripts to be published, please also confirm that the transcripts do not contain any potentially identifying information (or let us know if the participants consented to having their personal details published and made publicly available). We consider the following details to be identifying information:

- Names, nicknames, and initials

- Age more specific than round numbers

- GPS coordinates, physical addresses, IP addresses, email addresses

- Information in small sample sizes (e.g. 40 students from X class in X year at X university)

- Specific dates (e.g. visit dates, interview dates)

- ID numbers

Or, if the participants DID NOT provide consent for these transcripts to be published:

- Provide a de-identified version of the data or excerpts of interview responses

- Provide information regarding how these transcripts can be accessed by researchers who meet the criteria for access to confidential data, including:

a) the grounds for restriction

b) the name of the ethics committee, Institutional Review Board, or third-party organization that is imposing sharing restrictions on the data

c) a non-author, institutional point of contact that is able to field data access queries, in the interest of maintaining long-term data accessibility.

d) Any relevant data set names, URLs, DOIs, etc. that an independent researcher would need in order to request your minimal data set.

For further information on sharing data that contains sensitive participant information, please see: https://journals.plos.org/plosone/s/data-availability#loc-human-research-participant-data-and-other-sensitive-data

If there are ethical, legal, or third-party restrictions upon your dataset, you must provide all of the following details (https://journals.plos.org/plosone/s/data-availability#loc-acceptable-data-access-restrictions):

1. A complete description of the dataset

2. The nature of the restrictions upon the data (ethical, legal, or owned by a third party) and the reasoning behind them

3. The full name of the body imposing the restrictions upon your dataset (ethics committee, institution, data access committee, etc)

4. If the data are owned by a third party, confirmation of whether the authors received any special privileges in accessing the data that other researchers would not have

5. Direct, non-author contact information (preferably email) for the body imposing the restrictions upon the data, to which data access requests can be sent

Reviewers' comments:

Reviewer's Responses to Questions

**Comments to the Author**

1. Is the manuscript technically sound, and do the data support the conclusions?

Reviewer #1: Partly

2. Has the statistical analysis been performed appropriately and rigorously? 

Reviewer #1: No

3. Have the authors made all data underlying the findings in their manuscript fully available?

Reviewer #1: Yes

4. Is the manuscript presented in an intelligible fashion and written in standard English?

Reviewer #1: Yes

5. Review Comments to the Author

Reviewer #1: The study was well written but I think there is a problem in the design. The authors said they used mixed methods design yet they emphasized only the focus group discussion and empirical ranking.. In the results section I saw tables and percentages., which they did not state how they got those tables and the percentages from the participant. They never mentioned that they used a questionnaire and yet it seems they actually used. In addition what type of mixed method design did they use. Furthermore, how di they use the data generated from both qualitative and quantitative to arrive at their decision that women see prescription and accessibility as a challenge. The authors need to show it , other wise it would look like it was a decision they took by themselves. Yet, there is a need for evidence in order to be able to inform the policy positively.

6. PLOS authors have the option to publish the peer review history of their article (what does this mean?). If published, this will include your full peer review and any attached files.

Reviewer #1: No

---

## [Author Response · Author response to Decision Letter 0]

7 Apr 2024

Response to reviewer comments: 

Reviewer #1: The study was well written but I think there is a problem in the design. The authors said they used mixed methods design yet they emphasized only the focus group discussion and empirical ranking. In the results section I saw tables and percentages., which they did not state how they got those tables and the percentages from the participant. They never mentioned that they used a questionnaire and yet it seems they actually used. In addition, what type of mixed method design did they use. 

Response: We described our approach as a mixed methods (qualitative and quantitative) approach as it incorporated both focus group discussions (FGDs) and an empirical ranking exercise to explore women’s preferences regarding Over-The-Counter (OTC) oral contraceptive pills (OCPs). The empirical ranking exercise occurred after the FGDs and included organically generated factors identified during the FGDs. The FGDs and ranking exercises were given equal priority when interpreting the results. This type of approach has been successfully employed and reported upon previously in several studies in the healthcare context [1-4]. The manuscript has been revised to more clearly justify this approach.

The data collection section (Section 2.2) mentioned that “In both phases, participants were given a survey questionnaire collecting socio-demographic information, and their current and past use of contraceptives.” The results of this questionnaire formed the basis of the respondent characteristics reported in Table 1 and 2. The manuscript has been revised to make this more clear.

Reviewer #1: Furthermore, how did they use the data generated from both qualitative and quantitative to arrive at their decision that women see prescription and accessibility as a challenge. The authors need to show it , otherwise it would look like it was a decision they took by themselves. Yet, there is a need for evidence to be able to inform the policy positively.

Response: Prescription and accessibility as a challenge arose in both the FGDs and through the empirical ranking exercise, where it was the third most important factor. We have revised Section 3.2.1 of the manuscript to include a direct quote from the FGD regarding prescriptions being a constraint to accessibility: “I know I've skipped it for a week at a time if I couldn’t get to the doctor.” FGD1, ID-2

We have also revised section 3.3 to flag that the requirement for a prescription, which was the third most important factor in the ranking exercise, was also reflected in the FGD1 and FGD2 discussions.

Response to the editor:

1. Please ensure that your manuscript meets PLOS ONE's style requirements, including those for file naming. The PLOS ONE style templates can be found at: https://journals.plos.org/plosone/s/submission-guidelines

Response: According to the file name requirements, the online supplementary appendices have been renamed as:

• S1 Appendix: COREQ checklist

• S2 Appendix: Major and minor themes with responses from the participants

• S3 Appendix: Mind map for over-the-counter accessibility

Response: The scholarship numbers are the following: Macquarie University Research Excellence Scholarship (MQRES), No. 20211229, and MUCHE Top Up Scholarship, No. 20203478

3. Thank you for stating the following financial disclosure: "This research is funded by scholarships, including the MQRES and MUCHE Top Up scholarships, with support from Macquarie University and five pharmaceutical companies: Amgen Australia, Janssen Australia, MSD Australia, Pfizer Australia, and Roche Australia." Please state what role the funders took in the study. If the funders had no role, please state: ""The funders had no role in study design, data collection and analysis, decision to publish, or preparation of the manuscript."" If this statement is not correct you must amend it as needed. Please include this amended Role of Funder statement in your cover letter; we will change the online submission form on your behalf.

Response: The following sentence has been added “The funders had no role in study design, data collection and analysis, decision to publish, or preparation of the manuscript.”

4. Thank you for stating the following in the Financial Disclosure section: "This research is funded by scholarships, including the MQRES and MUCHE Top Up scholarships, with support from Macquarie University and five pharmaceutical companies: Amgen Australia, Janssen Australia, MSD Australia, Pfizer Australia, and Roche Australia." We note that you received funding from a commercial source: Amgen Australia, Janssen Australia, MSD Australia, Pfizer Australia, and Roche Australia. Please provide an amended Competing Interests Statement that explicitly states this commercial funder, along with any other relevant declarations relating to employment, consultancy, patents, products in development, marketed products, etc. Within this Competing Interests Statement, please confirm that this does not alter your adherence to all PLOS ONE policies on sharing data and materials by including the following statement: ""This does not alter our adherence to PLOS ONE policies on sharing data and materials.” (as detailed online in our guide for authors http://journals.plos.org/plosone/s/competing-interests). If there are restrictions on sharing of data and/or materials, please state these. Please note that we cannot proceed with consideration of your article until this information has been declared. Please include your amended Competing Interests Statement within your cover letter. We will change the online submission form on your behalf.

Response: Please amend the Competing Interests Statement to: “This research is funded by scholarships, including the MQRES (No. 20211229) and MUCHE Top Up scholarships (No. 20203478), with support from Macquarie University and five pharmaceutical companies: Amgen Australia, Janssen Australia, MSD Australia, Pfizer Australia, and Roche Australia. The funders had no role in study design, data collection and analysis, decision to publish, or preparation of the manuscript. NG has provided consultancy services on reclassification, including regarding oral contraceptives, and led the New Zealand work to gain pharmacist-supply access to oral contraceptives. NG was a Board Member of the Pharmaceutical Society of New Zealand, which provides training on non-prescription oral contraceptives. The de-identified transcripts are unable to be shared as the focus group discussions contain sensitive topics, such as discussions on reproductive health and contraception, brand names of medicines, specific medical centre/pharmacy etc. Data requests can be sent to: ethics.secretariat@mq.edu.au.”

5. Thank you for stating the following in the Competing Interests section: "NG has provided consultancy services on reclassification, including regarding oral contraceptives, and led the New Zealand work to gain pharmacist-supply access to oral contraceptives. NG was a Board Member of the Pharmaceutical Society of New Zealand, which provides training on non-prescription oral contraceptives." Please confirm that this does not alter your adherence to all PLOS ONE policies on sharing data and materials, by including the following statement: ""This does not alter our adherence to PLOS ONE policies on sharing data and materials.” (as detailed online in our guide for authors http://journals.plos.org/plosone/s/competing-interests). If there are restrictions on sharing of data and/or materials, please state these. Please note that we cannot proceed with consideration of your article until this information has been declared. Please include your updated Competing Interests statement in your cover letter; we will change the online submission form on your behalf.

Response: Please see above.

6. We note that you have indicated that there are restrictions to data sharing for this study. PLOS only allows data to be available upon request if there are legal or ethical restrictions on sharing data publicly. or more information on unacceptable data access restrictions, please see http://journals.plos.org/plosone/s/data-availability#loc-unacceptable-data-access-restrictions. Before we proceed with your manuscript, please address the following prompts:

b) If there are no restrictions, please upload the minimal anonymized data set necessary to replicate your study findings to a stable, public repository and provide us with the relevant URLs, DOIs, or accession numbers. For a list of recommended repositories, please see: https://journals.plos.org/plosone/s/recommended-repositories. You also have the option of uploading the data as Supporting Information files, but we would recommend depositing data directly to a data repository if possible.

Response: Please amend Data Availability statement to: There are restrictions imposed by the Macquarie University Ethics Committee on sharing the de-identified transcripts because the focus group discussions contain sensitive topics, such as discussions on reproductive health and contraception, brand names of medicines, specific medical centers/pharmacies etc. Data requests can be sent to: ethics.secretariat@mq.edu.au.

7. PLOS requires an ORCID iD for the corresponding author in Editorial Manager on papers submitted after December 6th, 2016. Please ensure that you have an ORCID iD and that it is validated in Editorial Manager. To do this, go to ‘Update my Information’ (in the upper left-hand corner of the main menu), and click on the Fetch/Validate link next to the ORCID field. This will take you to the ORCID site and allow you to create a new iD or authenticate a pre-existing iD in Editorial Manager. Please see the following video for instructions on linking an ORCID iD to your Editorial Manager account: https://www.youtube.com/watch?v=_xcclfuvtxQ

Response: Done

Response: Captions to the following Supporting Information files have been added to the end of the manuscript:

• S1 Appendix: COREQ checklist

• S2 Appendix: Major and minor themes with responses from the participants

• S3 Appendix: Mind map for over-the-counter accessibility

9. We note that this data set consists of interview transcripts. Can you please confirm that all participants gave consent for interview transcript to be published?

If they DID provide consent for these transcripts to be published, please also confirm that the transcripts do not contain any potentially identifying information (or let us know if the participants consented to having their personal details published and made publicly available). We consider the following details to be identifying information:

- Names, nicknames, and initials

- Age more specific than round numbers

- GPS coordinates, physical addresses, IP addresses, email addresses

- Information in small sample sizes (e.g. 40 students from X class in X year at X university)

- Specific dates (e.g. visit dates, interview dates)

- ID numbers

Or, if the participants DID NOT provide consent for these transcripts to be published:

- Provide a de-identified version of the data or excerpts of interview responses

- Provide information regarding how these transcripts can be accessed by researchers who meet the criteria for access to confidential data, including:

a) the grounds for restriction

b) the name of the ethics committee, Institutional Review Board, or third-party organization that is imposing sharing restrictions on the data

c) a non-author, institutional point of contact that is able to field data access queries, in the interest of maintaining long-term data accessibility.

d) Any relevant data set names, URLs, DOIs, etc. that an independent researcher would need in order to request your minimal data set.

For further information on sharing data that contains sensitive participant information, please see: https://journals.plos.org/plosone/s/data-availability#loc-human-research-participant-data-and-other-sensitive-data

If there are ethical, legal, or third-party restrictions upon your dataset, you must provide all of the following details (https://journals.plos.org/plosone/s/data-availability#loc-acceptable-data-access-restrictions):

a) A complete description of the dataset

b) The nature of the restrictions upon the data (ethical, legal, or owned by a third party) and the reasoning behind them

c) The full name of the body imposing the restrictions upon your dataset (ethics committee, institution, data access committee, etc)

d) If the data are owned by a third party, confirmation of whether the authors received any special privileges in accessing the data that other researchers would not have

e) Direct, non-author contact information (preferably email) for the body imposing the restrictions upon the data, to which data access requests can be sent

Response:

a) A complete description of the dataset. The data contains: 

• Transcripts of focus group discussions

• Ranking of the most important factors while accessing contraceptives

• Survey responses regarding respondent characteristics

b) The nature of the restrictions upon the data (ethical, legal, or owned by a third party) and the reasoning behind them: There are restrictions imposed by the Macquarie University Ethics Committee on sharing the de-identified transcripts because the focus group discussions contain sensitive topics, such as discussions on reproductive health and contraception, brand names of medicines, specific medical centres/pharmacies etc. The participants gave consent to “Only the investigators will have access to collected data.

c) The full name of the body imposing the restrictions upon your dataset (ethics committee, institution, data access committee, etc): Human Research Ethics Committee (HREC) Macquarie University.

d) If the data are owned by a third party, confirmation of whether the authors received any special privileges in accessing the data that other researchers would not have: No, the data is not owned by third party and the authors did not receive any special privileges in accessing the data that other researchers would not have. 

e) Direct, non-author contact information (preferably email) for the body imposing the restrictions upon the data, to which data access requests can be sent: Data requests can be sent to: ethics.secretariat@mq.edu.au

Response: The entire reference list has been checked one by one to ensure that there is no retracted article in the reference list. However:

• One reference has been updated from “MacKay MK, P. Schmidt, et al. Making hormo

---

## [Editor Report · Decision Letter 1]

24 May 2024

A qualitative exploration of the over-the-counter availability of oral contraceptive pills in Australia

PONE-D-23-29682R1

Dear  Zobaida Ahmed Piu,

We’re pleased to inform you that your manuscript has been judged scientifically suitable for publication and will be formally accepted for publication once it meets all outstanding technical requirements.

Kind regards,

Dr. Oluwatosin Olu-Abiodun

Academic Editor

PLOS ONE
---

## [Editor Report · Acceptance letter]

30 May 2024

PONE-D-23-29682R1 

PLOS ONE

Dear Dr. Ahmed, 

I'm pleased to inform you that your manuscript has been deemed suitable for publication in PLOS ONE. Congratulations! Your manuscript is now being handed over to our production team.

Kind regards, 

on behalf of

Dr. Oluwatosin Oluwaseun Olu-Abiodun 

Academic Editor

PLOS ONE